# Power, Muscle, and Take-Off Asymmetry in Young Soccer Players

**DOI:** 10.3390/ijerph17176040

**Published:** 2020-08-19

**Authors:** Petr Bahenský, David Marko, Václav Bunc, Pavel Tlustý

**Affiliations:** 1Department of Sports Studies, Faculty of Education, University of South Bohemia, 371 15 České Budějovice, Czech Republic; David.Marko@seznam.cz; 2Sports Motor Skills Laboratory, Faculty of Sports, Physical Training and Education, Charles University, 165 52 Prague, Czech Republic; Bunc@ftvs.cuni.cz; 3Department of Mathematics, Faculty of Education, University of South Bohemia, 371 15 České Budějovice, Czech Republic; tlusty@pf.jcu.cz

**Keywords:** asymmetry, soccer, strength, youth, muscles

## Abstract

(1) Background: The objective of the study was to check the relationship between laterality, amount of muscle mass (MM), and selected strength parameters on lower extremities and assessment of asymmetry like a result of training. (2) Methods: The screened sample consisted of soccer players (*n* = 65, age = 16.0 ± 1.2 years). The legs were assessed for MM, height of reflection on a force plate, and power over 30 s Wingate anaerobic test (WAnT). The relationships between the individual parameters and age dependence were assessed using a correlation analysis. The differences between the dominant and non-dominant leg were assessed using the *t*-test. (3) Results: A relationship between the jump height and the mean 30 s power in WAnT (*r* = 0.375, *p* < 0.01) and between the amount of MM and the absolute power of the individual legs in WAnT (*r* = 0.695–0.832, *p* < 0.01) was proved. A relationship between the take-off force and the MM, or between the MM and the relative power during a velocity force load was not found. (4) Conclusions: The amount of MM in young soccer players does not affect take-off force or strength power in WAnT. The more specific the movement is, the lower the effect on the achieved power output of the concerned MM. Differences in the performance between the dominant and non-dominant leg decrease with duration of the training.

## 1. Introduction

Lateralization that developed during human evolution has an internal and external background. It is especially manifested in the preference for one upper or lower limb in relation to motor skills and it influences the execution of the movement as well as the achieved power output [1,2]. About 90% of the population prefer their upper right limb for both working and physical activities [3,4]. Only 25–45% of people prefer their lower right limb for realization, for example jumping [3]. The preference of the lower limb might be influenced by the need for higher cerebral activation compared to upper limb movement [5]. The distal position of the lower limb muscles may also play a role [6]. Laterality is also influenced by various factors of the human body (genetics) and the environment (e.g., birth stress, hormonal activity) that may affect its form, even in the early postnatal period [7]. Structural asymmetry is considerably higher in men than in women [8].

Laterality plays an important role in a lot of sports that significantly use one of the limbs during exercise (ice hockey, tennis, etc.) including soccer. There is a relationship and transfer between the laterality of the upper and lower limbs [9]. Considering the ever-increasing pace of the game, players who are able to play with both legs have an advantage [10,11]. Therefore, players who use both legs are more useful in the game and they are much more valuable for the team than players with one leg preference [12]. According to some studies, the ability of players to play with both legs increases considerably with the increasing level of players and the time of performed intense training [13]. Additionally, the differences in the engagement and effectiveness of the dominant and non-dominant leg in ice-hockey have been studied [14]. In soccer, there are studies with similar topics [15] that did not register any strength differences between the preferred and non-preferred lower limb in adult female soccer players. The occurrence of bilateral knee flexor asymmetries at higher angle speeds was an important finding. It was found that there is at least one force asymmetry in 50% of players (bilateral knee flexor). In addition, an obvious leg preference in mobilization and stabilization tasks was found both in soccer players and in those who do not play soccer [16]. Among elite soccer players lower limb preferences can differ according to playing position [17].

Walking and running as a part of soccer performance are basic physical activities; they have a bipedal character and thus do not promote the development of differences between the limbs. The morphology of the limbs is affected by the duration of which the individual performs the procedure with only one limb. This time in soccer is negligible compared to some other sports (tennis, javelin throw, hockey, etc.). Currently, basically the entire content of the training leads to even development of the preferred and non-preferred lower limb in soccer players [18]. When comparing the level of difference in soccer players at various performance levels, insignificant differences between the one-foot jump distance and between the speed of repeated one-foot jumps were found [19]. The higher the level of the athletes (including soccer players), the lower their preference for their dominant limb in specific skills [10]. Physical development during adolescence influences the level of players [20] and the improved physical performance may minimize the scope of asymmetry and thus improve injury prevention [2]. The pursuit of a movement and structural balance is important in sports from a medical perspective as well, as some asymmetries may increase the risk of injury [21].

With the exception of some internal organs, man is symmetrical due to development. Asymmetry is tolerable, to some extent the individual compensates for it. If it exceeds the tolerable limit, the probability of injury increases significantly. Therefore, there are reasons to support the symmetrical development of the individual in the vast majority of sports. With asymmetrical movement, imbalances arise. They are based on an asymmetry in the amount of muscle mass, with some muscles shortening and some muscles weakening. In the event of a difference in the amount of muscle mass (MM) between the limbs, one limb may be overloaded, resulting in greater fatigue that may increase the risk of tissue damage and injury [21,22].

In trained athletes with good speed predispositions, the force in the execution of a physical activity based on take-off may be a more important factor than speed [23]. The motor performance of legs assessed by the Bosco Jump Test and Wingate anaerobic test (WAnT) is influenced by the maturity level of the assessed individuals, namely leg muscle development [24]. Both tests (the two-foot Bosco test and WAnT) correlate with increasing age [25]. The magnitude of the force executed by the striated skeletal muscle usually depends on the cross-section of the muscle [26,27]. This is used in sports where training is aimed to increase the muscle cross-section and where maximal hypertrophy is desirable [27,28,29]. However, there are sports where hypertrophy is not the goal, such as tennis, but where, from the point of view of laterality, one-sided specific training is reflected in the volume and quality of muscle mass [30]. The force should always be assessed in a specific manifestation. There is no dependence between the amount of muscle mass (MM) and the physical manifestation in the entire scope of the executed load intensity in athletes where a skill element is dominant in the execution of the physical activity. The volume of MM is only a necessary, not a sufficient condition for the physical activity, and the magnitude of the force is only a condition to some extent [28,31,32].

The basic objective in soccer training is to achieve a specific physical skill, while maximal muscle cross-section or maximal hypertrophy are not the objective. This means achieving a cross-section of the concerned muscles that ensures the execution of the required physical activity, which is optimal, not the maximal increase in the muscle cross-section. In this case, nervous coordination-adaptation is dominant [31]. Unlike other sports such as tennis [30], the training load of soccer players does not contain such an amount of specific load of the dominant limb, but there is an effort to evenly engage both limbs [33]. Movement skills in soccer performance are essential. Neuromuscular adaptation in sports where it plays a dominant role (in sports that are more demanding on coordination—skill-based), requires the force training to focus specifically on cultivating crucial physical skills [34]. Jump is one of the most frequently occurring game acts of a soccer player. Therefore, strength training in soccer is also aimed at while achieving the highest strength performance—jump height [35,36]. There is a minimal muscle hypertrophy in the specifically focused strength training [27,32,37,38,39]. The jump height and its changes depend on gender, age, and performance increases with age [40]. Soccer training focuses on cultivating specific physical skills; it influences the preferred physical skills, and the development of fitness is not primary. The development of “general” physical skills has to reach a level that is essential for the execution of specific soccer performance [31]. In general, it is not true that a higher level of force and speed potential leads to higher specific soccer performance. There are also different requirements for players at different positions [41]. The objective of this study was to determine the connection between laterality, amount of MM, speed-force performance, and training duration (the age of players) in top-level adolescent soccer players, and also to determine whether laterality (limb preference) affects the amount of MM. One of the goals was also to assess asymmetry as a result of training. This could be important for muscle morphology and also in order to prevent muscle disabilities and thus for injury prevention and the selection of training methods.

This study will verify whether laterality affects the morphological (MM) and functional (strength) parameters of young soccer players.

## 2. Materials and Methods

### 2.1. Subjects

The screened sample consisted of young players of the highest national league club, members of the teams U16–U19 category (*n* = 65, age = 16.0 ± 1.2 years, body height = 179.0 ± 6.5 cm, body mass = 71.0 ± 8.9 kg). Concerning the playing positions, the following players were assessed: eight goalkeepers, 15 defenders, 19 midfielders, and 23 attackers. To be included in the set of participants the following criteria must be fulfilled: active players in national sports level, age 15–18 years, and good health throughout last 2 months, without injuries throughout last 6 months.

All participants had been playing soccer since the age of 10; their average training load ranged from 12 to 16 h per week. The testing took place in the Laboratory of Load Diagnostics at DPSS FE FEU at the beginning of the competition period. Participants were not injured or rehabilitating from injury at time of testing.

### 2.2. Measures and Design

The dominant and non-dominant leg and hand were set, date of birthday was determined, as well as their player position. Players marked their preferred lower limb for kicking and jumping (confirmed by the coach) and their preferred hand for writing. The laboratory evaluation started by body composition assessment. Five minutes after that, take-off was measured. WAnT was used about 5 min after the jump test.

The body composition, including the amount of MM of individual legs, was measured by way of a Tanita BC 418 MA (Tanita Corp., Tokyo, Japan), and data were assessed by firm’s software. All participants were measured in the same time period (3–5 p.m.), before training, at least 3 h after lunch, without caffeine, for at least 1 h without drinking. The water intake was controlled during at 24 h before the evaluation.

The take-off from the right and left leg was then measured on a take-off board (Lode, Groningen, The Netherlands), which consists of a digital timer (maximum response time 10 ms) connected to a resistive platform. The flight time of the subject during the jump was thus measured. Take-off was performed from a squat position (knee angle at 90%), hands on hips. This design is standardized for take-off from both legs [17]. The test–retest (the take-off from the right and left leg) analysis revealed a high level of reliability between the two testing sessions (*r* = 0.907).

The strength parameters were evaluated on an Excalibur sport bicycle ergometer (Lode, Groningen, The Netherlands). A 5-min warm-up on the bicycle was followed by a 30 s all-out test. The test was used to determine the maximum relative power in 1 s and 30 s in each leg.

The goal of present study was to assess the relationship, influence of laterality on preference of the lower limb, the amount of MM, jumping abilities, and strength abilities in top level youth soccer players. The size of the difference in amount of MM between the left and right lower limb and the differences in the motor performance between the dominant and non-dominant leg was determined. The relationship between muscles mass on each lower limb, the level of jumping force, and WAnT performance was investigated as well as differences between the preferred and non-preferred lower limb. The differences between players of different ages, positions, and preferred lower limbs were also investigated.

### 2.3. Statistical Analysis

The data are represented as the average and the standard deviation (SD), unless otherwise indicated. The measured data were not modified. Shapiro–Wilk test was used for normality calculation, and data showed normal distribution. Dependence of the amount of MM, take-off skills, and force skills at WAnT were assessed using a correlation analysis (Pearson’s correlation coefficient). Linear regressions were calculated between one-legged jump and WAnT performance (difference of values between individual limbs). When determining the difference in muscle mass between the individual limbs and its relationship to age, the difference in muscle mass was determined as a percentage. When evaluating the relationship between jump height and power in WAnT, the difference between the right and left limbs was always evaluated by setting the value for left lower limb at 100%. The value measured on the right lower limb was related to this value. Significance was set at the *p* < 0.05 level. The critical value of Pearson’s correlation coefficient for a one-tailed test with 65 variables at the level of *p* = 0.05 is 0.244; at the level of *p* = 0.01, it is 0.318. The differences in the groups of players by leg preference in the individual parameters and in the individual players were assessed using the t-test at the level of significance of *p* < 0.05. Graphically, the differences between the dominant and non-dominant leg in the individual players from the aspect of the amount of MM, jump height, and power at two WAnT intensities (1 s and 30 s) was expressed. The differences were also assessed with regard to the age of the players, or the duration of their athletic career. Data processing was performed in Microsoft Excel (Microsoft, Redmond, WA, USA) and Statistica 12 (TIBCO Software Inc., Palo Alto, CA, USA).

### 2.4. Ethical Approval

All participants, or parents in case of minors, signed an informed consent form. The research was carried out with consent of the Ethics Committee, Faculty of Education, University of South Bohemia, Ref. No.: 002/2018. All procedures performed in the study were in accordance with the ethical standards of the institutional research committee and with the Helsinki declaration.

## 3. Results

Figure 1 shows the ratio in the upper and lower limb preferences. There is expressed laterality of players by which hand they write and which one is the preferred leg for kicking and for jumping in terms of playing soccer. The set of participants has a prevalence of individuals with a dominant right hand (for writing) and dominant right leg (for kicking) and dominant left leg (for take-off) (see Figure 1). As Figure 1 shows, 90% of the players with right hand preference prefer the right leg for kicking and 27% prefer the right leg for take-off. In total, 46% of players with left hand preference prefer the left leg for kicking and 54% prefer the left leg for take-off. Thus, there is a difference in lower limb preference between right-handed and left-handed players.

Table 1 presents the basic somatic, take-off parameters of the tested players and results during WAnT. The players show a higher volume of MM on the right leg than on the left (*p* < 0.01), regardless of their leg preference. There are no significant differences in the monitored parameters (amount of MM, jump height, engaging legs in WAnT) among the groups of players according to the leg preference for kicking and for take-off.

Figure 2 shows the distribution of players by their positions and preferred leg; 11 of them (two players on the left, eight in the center, and one on the right) are able to use both legs in the game comparably, according to the trainer’s assessment. It can be observed that the most represented profile was the right leg. Nobody from the attackers and goalkeepers prefers the left leg for kicking. Among defenders there are 32% of players with preferred left leg and among midfielders 24% of players.

Table 2 presents relationship between several pairs of parameters (amount of MM, WAnT performance, and jump height—on each lower limb).

Figure 3 and Table 2 present the relationship between the 30 s WAnT results and the jump height from the aspect of differences in the power outputs between the individual legs. A statistically significant dependence between the test results was determined (*p* < 0.01; *r* = 0.375).

Figure 4 presents an insignificant dependence (*p* = 0.26; *r* = 0.164) of the differences in power outputs of the individual legs on the age.

Figure 5 states the difference in the percentage amount of MM in the individual legs depending on the age. No dependence or trend was confirmed. No change in the differences of amount MM between the individual legs in the set of participants was registered.

Figure 6 shows the change in amount of MM during adolescence. The dependence of amount of MM in the legs on age is significant (*p* < 0.01; *r* = 0.399), and the amount of MM grows with age during adolescence.

The right leg was completely dominant in all monitored parameters in more than one-third of players. The distribution of the monitored variances was similar when the players were distributed by the preferred kicking leg. The opposite case, i.e., the left leg being better in all monitored indicators than the right leg, did not occur in our set, not even among the players who prefer their left leg both for kicking and take-off. On the contrary, two of those players had a completely dominant right leg. Furthermore, the results indicate that some players show a power imbalance in the monitored parameters (the largest difference in the power output of the individual legs was 33%), while the differences between the measured indicators for the right and left leg were low in others (at least 0%). The difference in the amount of MM between the left and right leg was 3.81% ± 1.68%; it was 7.07% ± 4.29% in 30 s WAnT, and 10.79% ± 7.85% in the one-foot jump.

No significant differences were demonstrated in the amount of MM, jump height, and power in WAnT between players with a different take-off leg (*p* = 0.82), or with a different preferred leg during the game (*p* = 0.85) or according to the preferred hand. It is therefore obvious that as far as laterality is concerned, there are no significant differences in the amount of MM, one-foot jump height, and power in WAnT between the players with left and right take-off legs.

## 4. Discussion

The most relevant results found the fact that laterality does not significantly affect the morphology and performance of soccer players. In the observed group of adolescents, there are no significant differences in the amount of MM, jump height, and power in WAnT between the groups of players with left and right leg preference for kicking and take-off. It was confirmed by the insignificant values of correlation coefficients. There are no significant differences among the position groups of players as well. The analysis of the game shows that the preferred foot of soccer players is unilaterally loaded only about 2% or less of the total playing time [42]. Thus, youth soccer training does not affect laterality. Neuromuscular coordination affects performance in terms of laterality [31].

The study did not find any relationship between the lower limb strength at WAnT and the amount of MM. The muscle strength and the amount of MM had no relationship to the dominance of the lower limb (take-off leg, kicking leg). No significant differences were registered in the take-off from the dominant and non-dominant leg, which is consistent with the published results [43].

It was also determined that the amount of MM increases considerably with age (by about 15.5% between the ages of 14 and 18), it is natural biological development, maturation. In this research the difference in the amount of MM between the dominant and non-dominant lower limb decreases insignificantly as age increases (by about 0.1% between the ages of 14 and 18). The difference between the dominant and non-dominant lower limb at maximal power in WAnT decreases by about 3% in the course of the monitored time period, which is significantly more than in the amount of MM. That is likely the consequence of the prevailing training locomotion which is specific for bipedal character. This suggests that the choice of training methods in the monitored players contributes to the symmetrical development of the body. This should help prevent asymmetries and reduce the likelihood of injury.

Another cause could be the higher effect of learning at a younger age and gradual loading of the training effect over the course of time [44]. These authors also confirmed a relationship between the intensity of conditioning training and improvement in physical skills. Balanced training, which puts an even load on the dominant and non-dominant limbs, may contribute to reducing potential differences in the engagement of limbs when executing a physical performance and to the development of required physical skills [45].

In case of one lower limb, power in WAnT is significantly influenced by the amount of MM. It was confirmed on the right lower limb and on the left one as well on both lower extremities—dominated and non-dominated or preferred or non-preferred. A significant relationship between the amount of MM and the absolute force power in the individual lower limbs was found. On the contrary, a relationship between amount of MM and the relative strength performance (converted to 1 kg weight of individual) related to the weight body mass of the individual lower limbs was not found. However, the differences in the power outcome in WAnT between the right and left lower limbs do not correspond to the differences in the amount of muscle mass, as well as the differences in jumps between the right and left lower limbs do not correspond to the differences in the amount of muscle mass. Thus, no general relationship between the amount of MM and muscle power was confirmed. This indicates the likely effect of the skill component.

The effect of laterality on the execution of strength performance is the subject of studies at various departments, as it may have a considerable influence on physical activity executed using only one limb. There are also other studies that deal with the topic of determining the preferred and non-preferred lower limb in soccer players and the differences in their composition and strength dispositions; for example, a relationship between the results of physiological tests and soccer performance was found [36]. Some studies deal with the relationship between the amount of MM and strength dispositions [26,31], others deal with relationships between the amount of MM and jump height [35]. The specificity of the training stimulus and its effect on strength development was additionally confirmed [37]. Significant differences in muscle activation between the dominant and non-dominant limb have not been confirmed neither in athletes nor non-athletes [34]. There are only a few studies that deal with the issue of the relationship between the lower limb amount of MM and speed-force performance, such as jump, in adolescents [40,46]. However, they are based on different methodology, take-off from both legs [40] or repeated jumps [46], but with similar results.

Considering the fact that the first step of measures to increase strength preconditions is to increase the cross-sections of the concerned muscles, MM hypertrophy, our study concentrated on an assessment of the relationship between the amount of MM and speed force of lower limbs, as well as the take-off force in adolescent soccer players.

Our study set included players with a dominant left limb (17%) and a dominant right limb (83%), comparable to published data on soccer players [47]. The ratio of players according to foot preference is similar for individual positions, except for attackers and goalkeepers. Due to the lack of players with a preferred left lower limb, these players are missing at some positions (left wing). This fact may influence the possibility of using tactical versions of the game. The take-off leg of the players is 70% left and 30% right, which is also consistent with published studies [3,47]. Most players (71%) have a different take-off leg and preferred leg in a specific performance—kicking. This might be caused by inherent dispositions for take-off, motion control and morphology of the concerned muscles (muscle fibers [3]), for limb preference [47], development of skills, and the essence of basic physical activity in soccer—locomotion with both legs. It has been reported that most right-handed individuals prefer the right lower limb for manipulation (kick, smoothing sand) and the left lower limb for posture securing [9]. The opposite tendency is recorded in left-handed individuals. In the case of right-handers, the results were confirmed in our study as well, but not for left-handers.

The determined significant relationship between the one-foot jump and 1 s power output of the same leg at WAnT means that the strength performance of the legs can be assessed in different ways. In practice, it implies options in the assessment of lower limb strength capacity in soccer players, both in jump and WAnT. However, due to the required equipment, WAnT is not usually available in field conditions. That corresponds to the already published conclusions [36] that have confirmed a close relationship between the maximum force (1 s WAnT) and jump height. The same was confirmed in volleyball players [25], while this dependence is closer in adults than in adolescents. It confirms the relationship between explosive force and take-off. Even closer dependence in the jump height difference between the dominant and non-dominant leg when executing a one-foot jump using individual legs and the difference in the power of the individual legs at 30 s WAnT was confirmed. A similar study was performed on healthy men and women between the ages of 17 and 20 who did not participate in any regular sports [48].

The limitations of our study include number of relatively homogeneous participants, the narrow age range of participants, different training history of players, and different capability of participants to learn new movement especially during the take-off from the non-dominant leg. However, the screened sample includes all soccer players from this region who fulfilled set conditions.

It shows that the amount of MM in adolescent soccer players is only a necessary, albeit insufficient, condition for strength performance, when it is assessed using movements that are difficult with respect to coordination [31,32,34]. The more general the movement is, or the more the individual is adapted to the specific physical activity, the closer the relationship between the amount of MM of the concerned muscle groups and the speed-force performance. This means that the amount of MM plays a significantly greater role in the execution of speed-force power in movements that the individual has adapted to than in physical activities with a specific character that are trained in a considerably shorter time period than basic physical activities [26]. The practical implications of the aforesaid are that it is essential to adjust the currently used conditioning methods in sports, the performance of which includes take-off and speed force.

## 5. Conclusions

This study confirmed a relationship between the take-off force and the explosive force of a single limb, as well as between the take-off force and the speed force of the limbs. There is a relationship between the amount of MM and the absolute power at WAnT. It was also confirmed that there is no relationship between take-off force and the amount of MM, or between the amount of MM and relative power at WAnT. That is an important factor when selecting conditioning methods. The laterality does not affect the morphology and performance of soccer players. The differences between the lower limbs decrease during the athletic career with the use of a sufficient and balanced load that respects individuality. Compensatory exercises are important for harmonious development, especially in adolescence.

## Figures and Tables

**Figure 1 ijerph-17-06040-f001:**
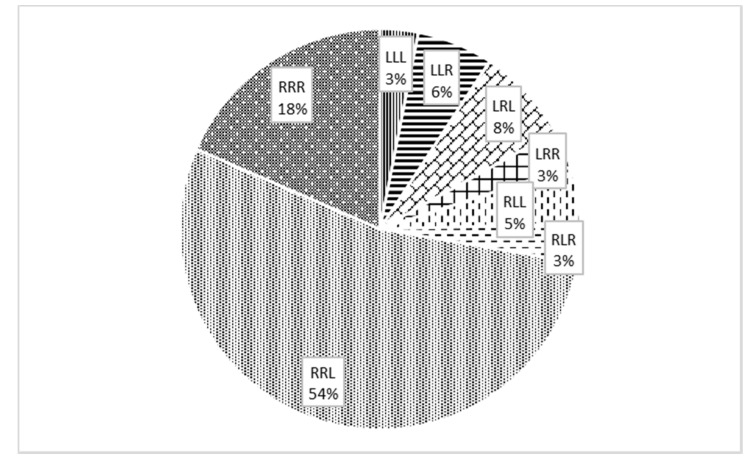
Percentage of representation of players by the preferred upper limb (writing), preferred lower limb (kicking) and take-off lower limb. 1st letter: which hand they use (L—left, R—right), 2nd letter: which leg they use for kicking, 3rd letter: which leg they use for take-off.

**Figure 2 ijerph-17-06040-f002:**
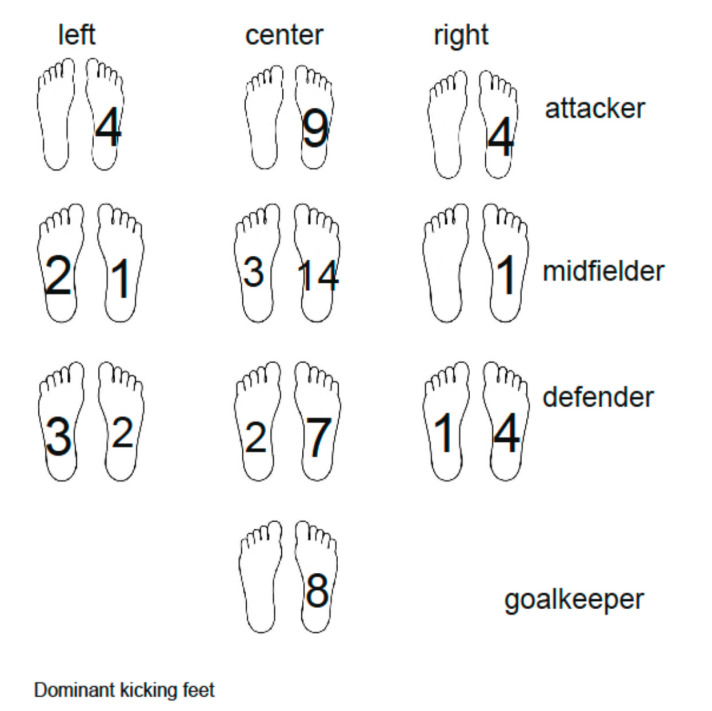
Dominant kicking leg—number of players by the position.

**Figure 3 ijerph-17-06040-f003:**
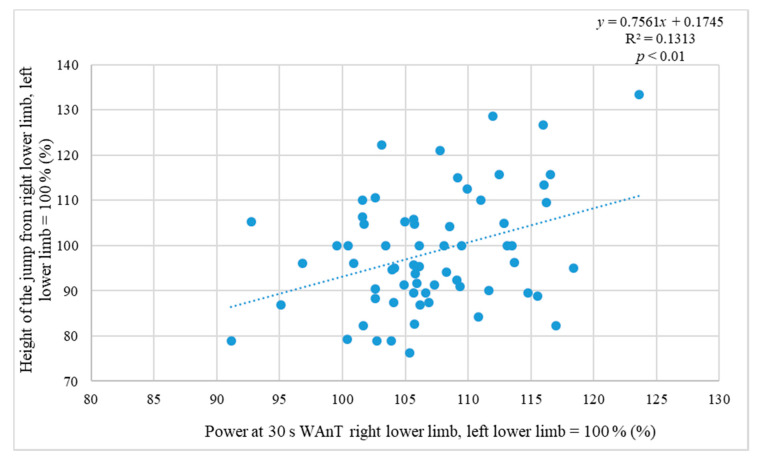
Relation of power at 30 s Wingate anaerobic test (WAnT) and jump height.

**Figure 4 ijerph-17-06040-f004:**
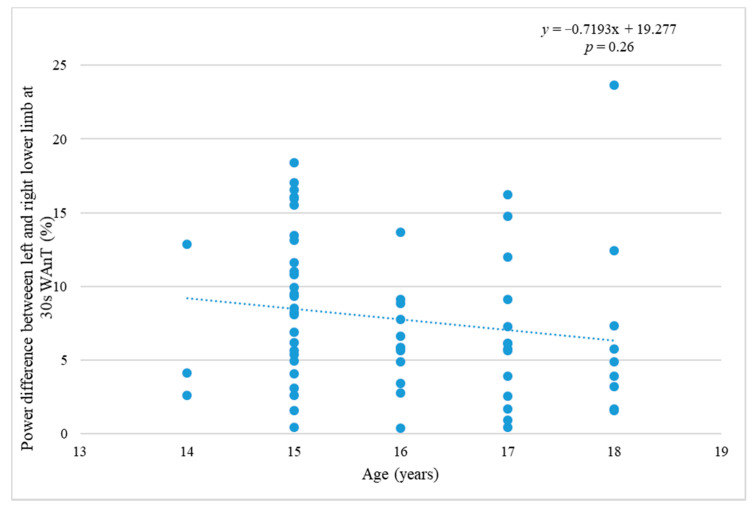
Relation of age and the power difference of individual legs at 30 s WAnT.

**Figure 5 ijerph-17-06040-f005:**
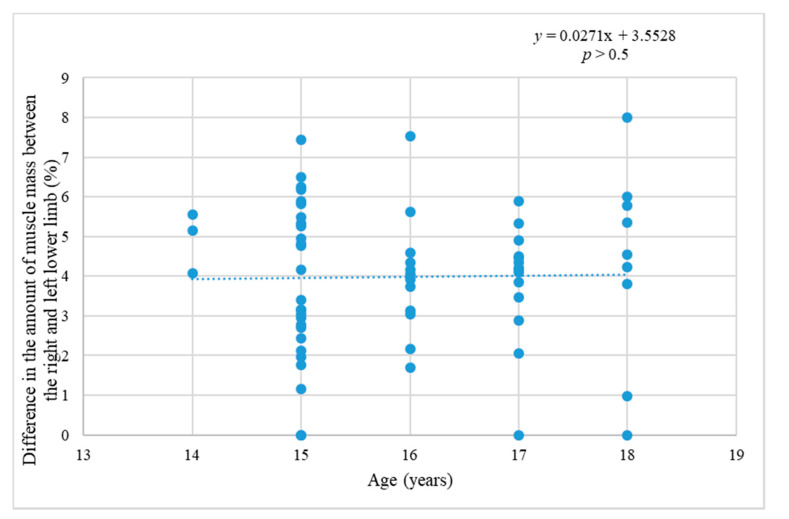
The relation of % difference in the amount of muscle mass (MM) in the individual legs and age.

**Figure 6 ijerph-17-06040-f006:**
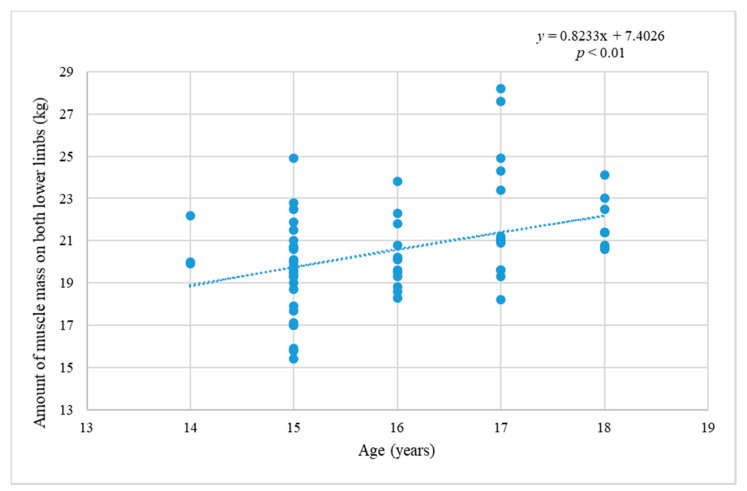
The relationship between the amount of MM in the both legs and age.

**Table 1 ijerph-17-06040-t001:** Measured values of the monitored participants by lower limb preference for kicking and by lower limb preference for take-off.

Parameters	Left Kicking	Right Kicking	Left Take-Off	Right Take-Off
Number of players (*n*)	11	54	45	20
Height (cm)	179.4 ± 7.9	178.9 ± 6.2	179.7 ± 6.7	177.4 ± 5.7
Body mass (kg)	69.3 ± 7.5	71.3 ± 9.1	71.1 ± 9.0	70.5 ± 8.6
Age (years)	16.5 ± 1.2	15.9 ± 1.1	15.9 ± 1.2	16.1 ± 1.1
MM right leg (kg)	10.09 ± 1.10	10.56 ± 1.26	10.52 ± 1.18	10.40 ± 1.38
MM left leg (kg)	9.86 ± 1.02	10.13 ± 1.22	10.10 ± 1.16	10.04 ± 1.27
Jump height right leg (m)	0.19 ± 0.04	0.19 ± 0.03	0.19 ± 0.03	0.18 ± 0.03
Jump height left leg (m)	0.18 ± 0.04	0.20 ± 0.03	0.20 ± 0.03	0.19 ± 0.03
1 s power of right leg (W)	1470.9 ± 161.8	1502.7 ± 221.8	1498.8 ± 221.7	1494.0 ± 192.7
1 s power of left leg (W)	1435.5 ± 133.2	1429.1 ± 224.4	1434.8 ± 225.2	1419.9 ± 177.3
30 s power of right leg (W)	1095.1 ± 121.4	1124.3 ± 156.2	1120.8 ± 157.9	1116.2 ± 135.0
30 s power of left leg (W)	1019.9 ± 94.4	1053.3 ± 143.6	1047.9 ± 136.5	1047.2 ± 138.5

Note: Differences in means of all followed variables are not statistically significant (*p* > 0.05).

**Table 2 ijerph-17-06040-t002:** Relationship between the individual variables.

Parameters	Correlation Coefficient	Significance
Difference R and L leg: jump height and power at 1 s WAnT	0.277	*p* < 0.025
Difference R and L leg: jump height and power at 30 s WAnT	0.375	*p* < 0.01
MM R leg and power at 30 s WAnT R leg	0.832	*p* < 0.001
MM L leg and power at 30 s WAnT L leg	0.811	*p* < 0.001
MM R leg and power at 1 s WAnT R leg	0.753	*p* < 0.001
MM L leg and power at 1 s WAnT L leg	0.695	*p* < 0.001
Difference R and L leg: MM and absolute power at 30 s WAnT	0.022	*p* > 0.50
Difference R and L leg: MM and relative power at 30 s WAnT	0.002	*p* > 0.50
Difference R and L leg: MM and absolute power at 1 s WAnT	0.082	*p* > 0.50
Difference R and L leg: MM and relative power at 1 s WAnT	0.035	*p* > 0.50
Difference R and L leg: MM and jump height	0.063	*p* > 0.50

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
