# Peer review of "Power, Muscle, and Take-Off Asymmetry in Young Soccer Players"

_ijerph, 2020, doi:10.3390/ijerph17176040_

Round 1
Reviewer 1 Report
Strength, Muscle and Take-off Asymmetry in Young Soccer Players
This study aimed to to check the effect of laterality on the relationship between the muscle mass (MM) with strength parameters. However, the design of this study does not allow us to check the effect of laterality. In addition, the conclusion is not supported by the results.
Introduction.
Pag. 1 Line 29, What authors mean by “during evolution” ?
Pag. 1 Line 35-36, authors mentioned that laterality is influenced by various factors of the environment, but we do not know what factors are.
Pag. 1 Line 38-38, the two postulates mentioned looks disconnected and we don’t know why authors are referring that, since we this paper talks about Young Soccer Players.
Pag. 2 Line 71-72 – the two sentences about the relationships with age should be merged.
Generally, the introduction is dispersed and touch on different points without enough deep. A better rationale for this study is needed.
Some important answers are missing, such as, what does this study add? What is missing about this topic? And how this study covers the main gap in the existing literature?
Materials and Methods
Pag. 3 Line 103; I assume that “Subjects” should be deleted.
Is there any rationale for the selected number of goalkeepers, defenders, midfielders, and attackers?
Is there any sample calculation estimation for this study?
Pag. 3 Lines 108. – 111; this information is repeated in the “2.5. Ethical Approval”
There was any inclusion criteria?
More information about take-off from the right and left leg measured on a take-off board is need.
More information about the protocols is needed. “Lode, Groningen, The Netherlands”. Is this a protocol? Is there any reference supporting this protocol? Are these tests validated?
Pag. 3 Lines 131. – 138; the information is repeated.
Pag. 4 Lines 145; revise.
Who performed the assessments? Was there any quality data control? Pilot study?
The Statistical analysis needs a better organization and explanations. Is not clear for why authors are performing this king of Statistical tests.
Is dependence assessed using a regression correlation analysis? What does it mean?
Results
Figure 1 is not the best way to present this data. It is a little bit confusing.
Pag. 5 Line 173. Revised
Introduce the p value on the table 1.
Is there any description of table 2 on the text?
Generally, the results need a better clarification. The statistical analysis performed need a better explanation to better understand the results.
Discussion
Please start the discussion by providing specific information about this study. For example, the most relevant results found.
Pag 8 line 249-250; “Functional differences between the dominant and non-dominant limb in athletes and non-athletes have also been confirmed”. The reference cited do not support this affirmation. Please revise this major point.
Pag. 9 line 250-252; There is several studies that deal with the issue of the relationship between lower limb MM and jump in adolescents.
Is there any limitation of this study to acknowledge?
The results do not allow the conclusion made in this study. It is hard to establish a link between the objectives and the conclusion of the study.
The discussion is also too generally a weak considering the dispersion and different populations cited to compare results.
An upgrade to the references is needed since there are several recently published studies on the topic.
Author Response
Dear reviewer,
thank you for your comments. Our notes are attached.

Reviewer 2 Report
This is interesting research, but there are some points that need to be addressed to could consider.
Introduction
Provide a concise rationale of the aim of the study, why is this important, clarified the objective? The introduction did not provide the basis of any research objective. In addition, this is not established
Methods
The methodology is quite a week, there was not described the BIA procedure used, there is no circumferences of legs muscle measured, no chronological age and maturation of the subject presented, no strength manifestation measurements...
The BIA used is quite basic, provide the validation study of this system
Why measure anaerobic power output? Why not strength manifestation, that is the factor referred in the title
Ethical approval is repeated in the method section
Results
Correctly presented
Discussion
Taking into account the lack of a clear objective is difficult to follow the discussion of the data obtained aiming to reach a concise conclusion that response to the original objective
Author Response

(The authors gave the same response as above.)

Round 2
Reviewer 1 Report
Abstract:
Since this is a descriptive study it is not possible to study the effect of laterality. Rather than authors can study the relationship between laterality, MM and strength parameters.
Introduction
“These factors can influence laterality: e.g., birth stress, hormonal activity, …” It looks authors forgot to complete this sentence.
This question need to be better address.
Is there hormonal activity, an environment factor?
Generally, the introduction keeps dispersed and touch on different points without enough deep. An effort is necessary to better explain how this study covers the main gap in the existing literature.
At the end of the introduction, the authors stress the importance of study this topic related to the prevention of muscle disabilities and injury prevention. This point is very important and should be better explain across the introduction.
Is there any sample calculation estimation for this study?
Pag. 3 Line 131-132 – please revise the sentence.
Pag.3 138-139 – is this the purpose of the study? Probably it is out of place.
Can you present the study design?
Pag. 4 line: 165 - Figure 1 shows the association between upper and lower limb preferences. It is not possible to see any association or relationship in this figure between upper and lower limb preferences.
“Differences of all the parameters are insignificant (p ˃ 0.05)”. Suggestion: differences in means are not statistically significant (p ˃ 0.05)”.
Pag. 8 Line 234 – 235 – With this study design, we cannot drive this cause-effect of age.
Pag. 8 Line 255-256: please revise the sentence.
Discussion: Pag. 9 Line “The authors did not find”- suggestion – This study did not find
Pag. 9 Line 266-267 – if this conclusion is based on this study, the authors should be careful because the study design does not allow them to take this conclusion. Probably longitudinal data is needed.
In the Discussion, the authors start saying that “There are no significant differences in amount of MM, jump height and power in WAnT between the groups of players”. However, in pag. 9 Line 278, they say that “Power in WAnT is significantly influenced by amount of MM”. It seems a little bit confused.
Generally, a better organization of the discussion, as well as to answer the objectives of this paper can improve the manuscript. The problem of increase the risk of injury in asymmetries players is not totally addressed in the discussion.
Reviewer 2 Report
Congratulation
Author Response
Response to Reviewer 2 Comments
Point 1: English language and style are fine/minor spell check required
Response 1: English language has been checked by native speaker.
Thank you for your revisions and recommendations.